# SNP-SNP positive interaction between MMP2 and MMP12 increases the risk of COPD

**Chimedlkhamsuren Ganbold[1], Jambaldorj Jamiyansuren** [ID][1]*, **Enkhbileg Munkhzorig[1]¤, Ichinnorov Dashtseren[2], Sarantuya Jav[1,3]**

**1** Department of Molecular Biology and Genetics, School of Biomedicine, Mongolian National University of Medical Sciences, Ulaanbaatar City, Mongolia, **2** Department of Pulmonology and Allergology, School of Medicine, Mongolian National University of Medical Sciences, Ulaanbaatar City, Mongolia, **3** Institute of Biomedical Science, Mongolian National University of Medical Sciences, Ulaanbaatar City, Mongolia

¤ Current address: Department of Basic Medical Sciences, School of Nursing, Mongolian National University of Medical Sciences, Ulaanbaatar City, Mongolia
* jambaldorj@mnums.edu.mn

**Data Availability Statement:** All relevant data are within the paper and its Supporting Information files.

**Funding:** This research received assistance from the Department of Science and Technology,

## Abstract

Determining SNP-SNP interaction of the disease has become important for further investigation of pathogenesis and experimental research. Although many studies have been published on the effect of *MMPs* gene polymorphisms on chronic obstructive pulmonary disease (COPD), there is a lack of information on SNP-SNP and SNP-environment interactions. This study aimed to investigate the interaction between the polymorphisms of *MMP1*, *MMP2*, *MMP9* and *MMP12* genes and its combined effect with smoking on the risk of developing COPD. Totally 181 COPD patients and 292 healthy individuals were involved. Blood samples from the participants were tested for genotyping and data were collected through questionnaires. Genotyping was performed with nested allele-specific polymerase chain reaction (AS-PCR) and polymerase chain reaction-restriction fragment length polymorphism (PCR-RFLP). SNP-SNP and SNP-environment interactions were investigated using multifactor dimensionality reduction and logistic regression analysis. The result showed that participants with high nicotine dependence and heavy smokers had a higher risk of COPD than non-smokers. Also, G/G genotype (cOR = 5.83; 95% CI, 1.19–28.4, p = 0.029) of *MMP2* rs243864 and T/T genotype (cOR = 1.79; 95% CI, 1.16–2.76, p = 0.008) of *MMP12* rs652438 independently contributes to the susceptibility of COPD. For SNP-SNP interaction, the positive interaction between rs243864 G/G genotype of *MMP2* and rs652438 T/T genotype of *MMP12* was found, and the combination of risk genotypes has a high risk of COPD (OR = 12.92; 95% CI, 1.46–114.4, p = 0.021). Moreover, the combination of T/T genotype of *MMP12* rs652438 and smoking-related factors increases the risk of COPD approximately 4.5 to 6-fold. The results suggests that there is a combination of *MMP2*, *MMP12*, and smoking-related factors may increase the risk of developing COPD.

Mongolian National University of Medical Sciences (Grant No.1/18), and the Mongolian Foundation for Science and Technology, Ministry of Science and Education, Mongolia (Grant no. Ш-2020/42). The funders had no role in study design, data collection and analysis, decision to publish, or preparation of the manuscript

**Competing interests:** The authors have declared that no competing interests exist.

## Introduction

Worldwide, a smoking is a common environmental risk factor for COPD [1]. It causes smoking-induced oxidative stress, inflammation and imbalance between protease and anti-protease system, leading to airway tissue destruction, abnormal remodeling, hypersecretion, fibrosis and airflow obstruction [2]. But only about 20 percent of heavy smokers had developed COPD [3]. It is mostly explained by an individual's genetic factors, which are related and contributes to detoxification, oxidative stress and other smoking-induced inflammation process [4]. Moreover, a number of studies have found the evidence for the association between smoking and matrix metalloproteinases (MMPs) in COPD [5–10]. During the pathogenesis of COPD with smoking-induced inflammation, MMPs play an important role in alveolar destruction and tissue remodeling processes [11]. Besides the proteolytic effect, studies showed that MMPs have an upregulating effect on pro-inflammatory factors [12].

Currently, twenty-three matrix metalloproteinases (MMPs) were determined that are expressed in human tissues. MMPs are classified by the structure and their substrates into collagenases (MMP1, MMP8, MMP13 and MMP18), gelatinases (MMP2 and MMP9), stromelysins (MMP3 and MMP10), matrilysins (MMP7 and MMP26), membrane anchored-type (MT-MMPs) and other MMPs (MMP12 and MMP28) [11]. The MMPs are major proteolytic enzymes involved in a variety of biological processes in humans, which degrades a variety of extracellular proteins, including collagens and elastin. Recent evidence implicates mostly the overexpression of MMPs leads to above mentioned physiological abnormalities associated with COPD [11, 13]. Hyeon-Kyoung Koo et al., found the plasma levels of MMP1 is associated with bronchodilator reversibility and emphysema index in COPD [13]. For MMP2, MMP9 gelatinases and MMP12, similar findings were observed. Serum levels of MMP2 were significantly higher in COPD patients compared with controls and asthmatic patients [14]. Robert Linder et al., found serum concentrations of MMP9 were increased among COPD patients and productive cough and decreased forced expiratory volume in the first second ($FEV_1$) were associated with it [15]. The overexpression and activation of MMPs linked to the genetic polymorphisms on its promotor region, NF-kB signaling, smoking-induced TLR4 activation, pro-inflammatory mediators, inhibition by tissue inhibitors of MMPs (TIMPs) and other signaling processes [16, 17]. Reports have found rs243864, rs3918242 and rs652438 polymorphisms, located on promotor region of *MMP2*, *MMP9* and *MMP12* genes, alters transcriptional activity and expression level of the protein [18–21]. Luis Santiago-Ruiz et al., reported that the participants with G/G genotype of rs11646643 polymorphism, had significantly higher plasma levels of MMP2 [22]. Also, rs1799750 polymorphism was associated with higher expression levels of MMP1 and COPD risk [23, 24]. On the other hand, rs243864, rs11646643 of *MMP2*, rs3918242, rs3918253 of *MMP9* and rs2276109 of *MMP12* polymorphisms and the interactions between them, were associated with COPD among Serbians and Mexicans [24, 25]. In the present study we compare the allelic and genotypic frequencies and their potential interactions of the polymorphisms with smoking-related factors between the groups and assess their risk for COPD among Mongolian population.

## Materials and methods

### Study population

Each participant in this study obtained the required information, agreed to participate voluntarily, and signed an informed consent form. The Research Ethics Committee of the Ministry of Health, Mongolia, and the Biomedical Research Committee of the Mongolian National University of Medical Sciences granted ethical approval to conduct this study. Between October

2016 and February 2019, 181 patients diagnosed with COPD who were under the supervision of the Pulmonology Department of the First, Second and Third State Hospitals of Mongolia were included in the case group. According to the Global Initiative for Chronic Obstructive Lung Disease (GOLD) diagnostic criteria, the inclusion criteria for the study are as follows: a chronic or recurrent cough or sputum production for 3 months, $\geq$40 age, an $FEV_1$ <70% of predicted, an $FEV_1$/forced vital capacity (FVC) ratio of <0.70 and an increase in $FEV_1$ of <12% 15 min after the inhalation of 400 µg Ventolin (albuterol sulfate) [26]. In the control group, 292 healthy individuals over 40 years of age, not genetically related to participants in the case group, and without a family history of COPD were included. All participants completed the COPD diagnostic questionnaire (CDQ) and Fagerström Test for Nicotine Dependence (FTND) questionnaires and spirometry test, which were performed according to the guidelines by trained professionals using the EasyOne Pro® (ndd Medical Technologies Inc., Switzerland). With the CDQ questionnaire, we collected information on smoking status, cigarette per day (CPD), years of smoking and age of smoking initiation (ASI). The pack-years was calculated as the number of packs smoked per day multiplied by the number of years for smoking. Ex-smokers were excluded from the study. We classified the current smokers by CPD as heavy smokers if they smoked 20 or more cigarettes per day or light smokers if they smoked less than 20 cigarettes per day. Using the FTND questionnaire, nicotine dependence was assessed on a scale of 0 to 10 and classified into low (0–3 scores) and high dependence (4–10 scores) categories [27].

## SNP genotyping

Nucleic acid was manually extracted from a blood sample using the DNeasy Blood and Tissue Kit (QIAGEN, Germany) according to the instruction and it's stored at -20˚C. We used nested AS-PCR to detect *MMP9* rs3918242 genotypes [28]. Also, RFLP was used to detect the variations of rs1799750 of *MMP1*, rs11646643 and rs243864 of *MMP2*, rs3918253 of *MMP9*, rs652438 of *MMP12* gene [29–31]. List of the primers, restriction enzymes and length of amplicons are shown in S1 Table. PCR reactions were performed using the AccuPower® Hot-Start PCR PreMix Kit (K-5050, Bioneer Corporation, Korea). The amplicons and their length were determined by agarose gel electrophoresis (C-9100-1, Bioneer Corporation, Korea) and visualized with ethidium-bromide staining (C-9036, Bioneer Corporation, Korea).

## Statistical analysis

Microsoft Excel (Microsoft Corporation, USA) and STATA 13.0 (StataCorp, USA) software are used for descriptive and inferential analyses. Age, body mass index (BMI), $FEV_1$, FVC, and $FEV_1$/FVC ratio were compared between case and control groups by using the Student's t-test, ANOVA, or Mann–Whitney U-test. For frequency of allele, genotype and other nominal variables, which were analyzed by A Pearson's chi-squared test ($\chi$2) and Fisher's exact test with 2x2, 2x3, or 2×4 contingency tables. A statistically significant difference was considered when the p-value was less than 0.05. The genetic model of the variations were determined by the computational method of four-model strategy described by Horita and Kaneko [32]. Using MDR 3.0.2 software, multifactor dimensionality reduction (MDR) analysis was performed to identify SNP-SNP interactions and the best model with the risk for COPD. The cross-validation consistency (CVC) was calculated by a 10-fold cross-validation procedure for reduce the chance of false positives. For choosing the best model, the model with the highest CVC, training balance accuracy (TrBA) and testing balance accuracy (TeBA) variables was selected. The relative excess risk due to interaction (RERI), synergy index (S) and attributable proportion (AP), were calculated to determine whether SNP-SNP or SNP-environment combinations had

positive or negative interactions [33]. If the RERI and AP values of 0 implies no interaction or exactly additivity whereas values higher than 0 imply positive interaction or more than additivity and values lower than 0 imply negative interaction or less than additivity. For the S value, it equals 1 means no interaction or exactly additivity whereas greater than 1, which means positive interaction or more than additivity and less than 0 means negative interaction or less than additivity. The gene network was constructed using GeneMANIA. The physical interactions, co-expression, predictions, co-localization, genetic interaction, pathway and shared protein domain networks were performed. Also, it showed the physiological functions that the selected genes and their proteins were involved. We used logistic regression analysis to calculate the crude (cOR) and adjusted odds ratio (aOR) with 95% confidence intervals (CI) to determine the association between the factors and COPD. The post-hoc test was performed to calculate the statistical power.

## Results

### The sociodemographic, smoking and spirometry variables of study participants

181 COPD patients and 292 controls participated in the present study. The comparison of sociodemographic, spirometric and smoking data between study groups is shown in Table 1. The age, gender, BMI, education, dust exposure and smoking period variables were similar; no differences were found between the groups. However, the ratio of current to never smokers, CPD, pack-years smoked and spirometry parameters were significantly different between COPD and control groups.

According to the categories of smoking status, pack year, CPD, ND and ASI variables, the groups were divided into subgroups to examine the differences between them. As shown in Table 2, current smokers (cOR = 1.68; 95% CI, 1.08–2.64, p = 0.02) and participants who had early onset of daily smoking (cOR = 2.19; 95% CI, 1.17–4.09, p = 0.014) had a higher risk of

**Table 1. Sociodemographic, smoking and spirometry variables of study groups.**

| Variables | Unit and Category | COPD (n = 181) | Controls (n = 292) | P value |
|---|---|---|---|---|
| Age | years / mean (95% CI) | 63.57 (62.16–64.98) | 63.23 (62.16–64.3) | 0.702[a] |
| Gender | male / n (%) | 119 (65.7%) | 175 (59.9%) | 0.205[b] |
| BMI | $kg/m^2$ / mean (95% CI) | 25.3 (24.95–25.65) | 25.48 (25.21–25.75) | 0.426[a] |
| Education | Primary / n (%) | 41 (22.7%) | 54 (18.5%) | 0.362[b] |
| | Secondary / n (%) | 39 (21.5%) | 77 (26.4%) | |
| | Tertiary / n (%) | 101 (55.8%) | 161 (55.1%) | |
| Dust exposure | yes / n (%) | 91 (50.3%) | 138 (47.3%) | 0.524[b] |
| Smoker | yes / n (%) | 146 (80.7%) | 208 (71.2%) | 0.022[b] |
| CPD | number / mean (95% CI) | 18.38 (16.99–19.77) | 14.94 (14.07–15.81) | <0.001[a] |
| Smoking period | years / mean (95% CI) | 34.63±13.17 | 32.43±11.82 | 0.087[a] |
| Pack-years | pack years / mean (95% CI) | 33.03 (29.81–36.25) | 25.05 (23.17–26.93) | <0.001[c] |
| Spirometry parameters | $FEV_1$ (L) / mean (95% CI) | 1.36 (1.28–1.44) | 2.86 (2.81–2.91) | <0.001[c] |
| | $FEV_1$ (%) / mean (95% CI) | 48.41 (46.24–50.58) | 89.91 (89.08–90.74) | <0.001[c] |
| | FVC (L) / mean (95% CI) | 2.82 (2.69–2.95) | 3.59 (3.52–3.66) | <0.001[c] |
| | FVC (%) / mean (95% CI) | 77.03 (74.28–79.77) | 87.70 (86.49–88.91) | <0.001[c] |
| | $FEV_1$/FVC / mean (95% CI) | 0.49 (0.47–0.51) | 0.80 (0.79–0.81) | <0.001[c] |

BMI, Body mass index; CPD, Cigarette per day; $FEV_1$, forced expiratory volume in the first second; FVC, forced vital capacity.

P values were calculated by [a]Student's t-test, [b]Chi-squared ($x^2$) test or [c]Mann-Whitney U test.

**Table 2. Association between smoking-related variables and COPD risk.**

| Categories | COPD (n = 181) | Controls (n = 292) | cOR | 95%CI | Power | aOR | 95%CI | Power |
|---|---|---|---|---|---|---|---|---|
| Never smoker | 35 (19.3) | 84 (28.8) | 1 | - | - | 1 | - | - |
| Current smoker | 146 (80.7) | 208 (71.2) | 1.68* | 1.08–2.64 | 64.3% | 1.51 | 0.96–2.39 | 36.3% |
| CPD <20 | 71 (39.2) | 146 (50.0) | 1.17 | 0.72–1.89 | 63.1% | 1.06 | 0.65–1.74 | 35.8% |
| CPD ≥20 | 75 (41.5) | 62 (21.2) | 2.90*** | 1.73–4.88 | 99.7% | 2.87** | 1.67–4.94 | 97.9% |
| Low-ND | 55 (30.4) | 114 (39.0) | 1.16 | 0.69–1.93 | 47.5% | 1.07 | 0.63–1.81 | 22.1% |
| High-ND | 91 (50.3) | 94 (32.2) | 2.32*** | 1.43–3.79 | 97.5% | 2.18* | 1.32–3.59 | 87.8% |
| ASI >16 years | 115 (63.6) | 174 (59.6) | 1.59* | 1.00–2.51 | 13.6% | 1.47 | 0.92–2.35 | 3.6% |
| ASI ≤16 years | 31 (17.1) | 34 (11.6) | 2.19* | 1.17–4.09 | 39.7% | 2.01 | 1.04–3.89 | 17.7% |
| Pack-years <20 | 48 (26.5) | 101 (34.6) | 1.14 | 0.68–1.92 | 45.3% | 1.06 | 0.62–1.81 | 20.4% |
| 20–39 pack-years | 47 (26.0) | 64 (21.9) | 1.76* | 1.02–3.04 | 17.7% | 1.68 | 0.96–2.94 | 5.4% |
| Pack-years ≥40 | 51 (28.2) | 43 (14.7) | 2.85*** | 1.62–5.01 | 94% | 3.02** | 1.64–5.57 | 81.2% |

CPD, Cigarette per day; ND, Nicotine dependence; ASI, Age of smoking initiation; cOR, Crude odd's ratio; aOR, Adjusted odd's ratio; CI, confidence interval. The values were given as numbers (proportion).

P value by two-tailed Chi-squared ($x2$) test and it noted as

*p<0.05

**p<0.01 and

***p<0.001. aOR and confidence interval were calculated by logistic regression. Adjusted for age, gender, BMI, education, dust exposure and smoking years; Statistical power was calculated by post-hoc test.

COPD compared with never smokers. Significant differences were found for pack-years of smoking (cOR = 2.85; 95% CI, 1.62–5.01, p = 0.0002), ND (cOR = 2.32; 95% CI, 1.43–3.79, p = 0.0006) and CPD (cOR = 2.9; 95% CI, 1.73–4.88, p <0.001) between the groups. The heavy smokers (CPD ≥20, aOR = 2.87; 95% CI, 1.67–4.94, p = 0.002), participants who had over 40 pack-years (aOR = 3.02; 95% CI, 1.64–5.57, p = 0.004) and high nicotine dependence (aOR = 2.18; 95% CI, 1.32–3.59, p = 0.031), had a 2 to 3-fold higher risk for COPD compared with never-smokers by multivariate analysis.

## Alleles and genotypes of SNP polymorphisms

No significant differences between COPD patients and controls have been observed for allele frequencies of rs1799750, rs11646643, rs243864, rs3918253 and rs3918242. For rs652438 of *MMP12*, the T allele (cOR = 1.57; 95% CI, 1.07–2.30, p = 0.021) was more frequent among COPD patients than controls (Table 3).

On comparing the genotype frequency of *MMP2* rs243864, G/G homozygote genotype (G/G vs T/G + T/T, cOR = 5.83; 95% CI, 1.19–28.4, p = 0.029) and *MMP12* rs652438, T/T homozygote genotype (T/T vs T/C + C/C, cOR = 1.79; 95% CI, 1.16–2.76, p = 0.008) were significant to increased risk of COPD in recessive model. By multivariate analysis, the same significance was observed for rs652438 T/T (aOR = 1.68; 95% CI, 1.12–2.58, p = 0.012) and rs243864 G/G (aOR = 5.67; 95% CI, 1.25–23.2, p = 0.031) genotypes. Contrary, no significant difference was shown for genotype frequencies of rs1799750, rs11646643, rs3918253 and rs3918242 (Table 4).

## Gene–Gene and Gene–Smoking interactions

By the MDR analysis, best interaction models identified MDR from 10-fold cross-validation for COPD, are listed in Table 5. Statistically significance was observed for one (rs652438, $x^2$ = 7.53, p = 0.023, mdrOR = 1.87; 95% CI, 1.19–2.94), two (rs243864-rs652438, $x^2$ = 17.39, p = 0.015, mdrOR = 2.45; 95% CI, 1.44–4.15) and three (rs243864-rs3918242-rs652438, $x^2$ =

**Table 3. Comparison of allele frequencies between groups.**

| Gene | refSNP ID | Alleles | Risk allele | RAF | | cOR | 95%CI | Power |
|------|-----------|---------|-------------|-----|-----|-----|-------|-------|
| | | | | Cases (n = 362) | Controls (n = 584) | | | |
| MMP1 | rs1799750 | CC>delC | CC | 0.671 | 0.657 | 1.06 | 0.81–1.40 | 9.1% |
| MMP2 | rs11646643 | A>G | G | 0.304 | 0.291 | 1.06 | 0.79–1.42 | 6.3% |
| MMP2 | rs243864 | T>G | G | 0.16 | 0.123 | 1.36 | 0.93–1.97 | 41.8% |
| MMP9 | rs3918253 | C>T | T | 0.146 | 0.125 | 1.2 | 0.82–1.76 | 26.8% |
| MMP9 | rs3918242 | C>T | C | 0.831 | 0.813 | 1.13 | 0.80–1.59 | 11.5% |
| MMP12 | rs652438 | T>C | T | 0.881 | 0.825 | 1.57* | 1.07–2.30 | 70.1% |

RAF, Risk allele frequency; cOR, Crude odd's ratio; CI, confidence interval. The values were given as frequency. The crude odd's ratio, confidence interval and p values were calculated by logistic regression. For the significant difference, p value noted as

*p<0.05

**p<0.01 or

***p<0.001. Statistical power was calculated by post-hoc test.

35.41, p = 0.018, mdrOR = 2.14; 95% CI, 1.45–3.16) SNP-SNP interaction models by MDR analysis.

Further analysis of SNP-SNP interactions, found associations between one to three loci model and COPD risk. By two SNPs interaction model, we found positive interaction and the result showed carriers of rs243864 G/G of *MMP2* and rs652438 T/T of *MMP12*, had increased risk for COPD (OR = 12.92; 95% CI, 1.46–114.4, p = 0.021; RERI = 6.94; AP = 0.537; S = 2.395) compared with participants without any of the risk genotypes (Table 6). Through three loci interaction model, it showed that the risk of COPD significantly increases with the number of risk genotypes (OR = 12.0; 95% CI, 1.10–131.24, p = 0.042; RERI = 7.76; AP = 0.647; S = 3.396) compared with participants without any of these risk genotypes. No association and interactions were observed for the four SNP models.

The multiple gene association network from GeneMANIA, showed *MMP2* and *MMP12* were linked through co-expression (weight: 8.01%) and shared protein domains (weight: 0.6%). Also, *MMP2*, *MMP9* and *MMP12* were involved in extracellular matrix organization

**Table 4. Genotype frequencies of SNPs in selected genetic models among groups.**

| Gene | refSNP ID | Genetic model | Risk genotypes | Genotype frequency | | cOR | 95%CI | Power | aOR | 95%CI | Power |
|------|-----------|---------------|----------------|-------------------|-----|-----|-------|-------|-----|-------|-------|
| | | | | Cases (n = 181) | Controls (n = 292) | | | | | | |
| MMP1 | rs1799750 | Recessive | CC/CC | 0.40 | 0.377 | 1.12 | 0.77–1.64 | 9.6% | 1.04 | 0.69–1.73 | 7.3% |
| MMP2 | rs11646643 | Recessive | G/G | 0.09 | 0.05 | 1.91 | 0.93–3.93 | 40.7% | 1.75 | 0.72–3.97 | 37.3% |
| MMP2 | rs243864 | Recessive | G/G | 0.039 | 0.007 | 5.83* | 1.19–28.4 | 61.4% | 5.67* | 1.25–23.2 | 54.6% |
| MMP9 | rs3918253 | Recessive | T/T | 0.049 | 0.027 | 1.86 | 0.70–4.91 | 21.1% | 1.71 | 0.62–4.88 | 22.8% |
| MMP9 | rs3918242 | Recessive | C/C | 0.724 | 0.661 | 1.34 | 0.89–2.02 | 29.7% | 1.21 | 0.71–1.98 | 27.2% |
| MMP12 | rs652438 | Recessive | T/T | 0.79 | 0.678 | 1.79** | 1.16–2.76 | 76% | 1.68* | 1.12–2.58 | 72.9% |

cOR, Crude odd's ratio; aOR, Adjusted odd's ratio; CI, confidence interval. The values were given as frequency. The crude or adjusted odd's ratio, confidence interval, p values were calculated by logistic regression. For adjusted odd's ratio, it adjusted for age, gender, BMI, education, dust exposure and smoking years. Statistical power was calculated by post-hoc test. For the significant difference, p value noted as

*p<0.05

**p<0.01 or

***p<0.001. The genetic model was selected by four-model strategy described by Horita and Kaneko [32].

**Table 5. Best models of SNP-SNP interactions.**

| Models | Training Bal.Acc. | Testing Bal.Acc. | Sign test (p) | CVC | Chi-squared | mdrOR (95% CI) |
|---|---|---|---|---|---|---|
| rs652438 | 0.557 | 0.547 | 0.023 | 10/10 | 7.53 | 1.87 (1.19–2.94)* |
| rs243864+rs652438 | 0.565 | 0.5311 | 0.015 | 7/10 | 17.39 | 2.45 (1.44–4.15)* |
| rs243864+rs3918242+rs652438 | 0.596 | 0.491 | 0.018 | 4/10 | 35.41 | 2.14 (1.45–3.16)* |
| rs1799750+rs11646643+rs3918253+rs652438 | 0.641 | 0.579 | 0.046 | 10/10 | 60.93 | 3.14 (2.14–4.63) |
| rs1799750+rs11646643+rs3918253+ rs3918242+rs652438 | 0.676 | 0.541 | 0.159 | 7/10 | 91.47 | 4.13 (2.78–6.14) |
| rs1799750+rs11646643+rs243864+rs3918253+ rs3918242+rs652438 | 0.716 | 0.551 | 0.141 | 10/10 | 138.87 | 6.10 (4.03–9.22) |

Training Bal. Acc, Training Balanced Accuracy; Testing Bal. Acc, Testing Balanced Accuracy; CVC, Cross Validation Consistency; mdrOR, Overall odd's ratio; CI, confidence interval; Overall odd's ratio, confidence interval and p values were calculated by MDR; The best model speculated by MDR is composed of rs1799750, rs11646643, rs243864, rs3918253, rs3918242 and rs652438. For the significant difference, p value noted as

*p<0.05

**p<0.01 or

***p<0.001.

(FDR = 3.58e-29), collagen metabolic process (FDR = 5.62e-23) and metallopeptidase activity (FDR = 1.11e-14) (Fig 1).

As shown in Table 7, we found a positive interaction between the T/T genotype of *MMP12* and smoking-related factors. The risk of COPD was 6-fold higher for heavy smokers with the T/T genotype of *MMP12* than light smokers with the non T/T genotype (cOR = 6.21; 95% CI, 2.81–13.72, p<0.001, RERI = 0.671; AP = 0.108, S = 1.148). Also, T/T genotype positively interacted with high ND (cOR = 4.55; 95% CI, 2.04–10.13, p <0.001, RERI = 0.448; AP = 0.099,

**Table 6. The cumulative effect of best models of SNP-SNP interactions on COPD.**

| SNP x SNP | | | Cases (n = 181) | Controls (n = 292) | cOR | 95%CI | P value | aOR | 95%CI | P value |
|---|---|---|---|---|---|---|---|---|---|---|
| Two-way model | | *MMP12* (rs652438) | | | | RERI = 6.94; AP = 0.537; S = 2.395; | | | | |
| *MMP2* (rs243864) | Non G/G | Non T/T | 36 (0.199) | 93 (0.318) | 1 | - | - | 1 | - | - |
| | | T/T | 138 (0.762) | 197 (0.676) | 1.81 | 1.16–2.82 | 0.008 | 1.73 | 1.13–2.76 | 0.012 |
| | G/G | Non T/T | 2 (0.011) | 1 (0.003) | 5.17 | 0.45–58.75 | 0.185 | 6.09 | 0.39–63.21 | 0.251 |
| | | T/T | 5 (0.028) | 1 (0.003) | 12.92 | 1.46–114.4 | 0.021 | 11.89 | 1.21–98.52 | 0.002 |
| Three-way model (rs243864, rs3918242 and rs652438) | | | | | | | | | | |
| Number of risk genotypes | | | | | | RERI = 7.76; AP = 0.647; S = 3.396; | | | | |
| 0 | | | 8 (0.044) | 32 (0.109) | 1 | - | - | 1 | - | - |
| 1 | | | 68 (0.376) | 128 (0.439) | 2.13 | 0.93–4.87 | 0.075 | 2.03 | 0.79–4.91 | 0.083 |
| 2 | | | 102 (0.564) | 131 (0.449) | 3.11 | 1.38–7.05 | 0.006 | 2.99 | 1.08–7.27 | 0.018 |
| 3 | | | 3 (0.016) | 1 (0.003) | 12.0 | 1.10–131.24 | 0.042 | 11.2 | 1.03–128.74 | 0.048 |
| Four-way model (rs1799750, rs11646643, rs3918253 and rs652438) | | | | | | | | | | |
| Number of risk genotypes | | | | | | RERI = -1.104; AP = -0.479; S = 0.542; | | | | |
| 0 | | | 17 (0.127) | 49 (0.322) | 1 | - | - | 1 | - | - |
| 1 | | | 92 (0.348) | 169 (0.414) | 1.57 | 0.85–2.88 | 0.146 | 1.32 | 0.78–2.56 | 0.237 |
| 2 | | | 68 (0.287) | 69 (0.233) | 2.84 | 1.49–5.42 | 0.002 | 2.67 | 1.31–5.37 | 0.012 |
| 3 | | | 4 (0.183) | 5 (0.028) | 2.31 | 0.55–9.59 | 0.251 | 2.22 | 0.43–8.99 | 0.411 |

cOR, Crude odd's ratio; aOR, Adjusted odd's ratio; CI, confidence interval; RERI, relative excess risk due to interaction; AP, attributable proportion; S, synergy index. The values are given as numbers (frequency). Odd's ratio and confidence interval were calculated by logistic regression. Adjusted for age, gender, BMI, education, dust exposure and smoking status.

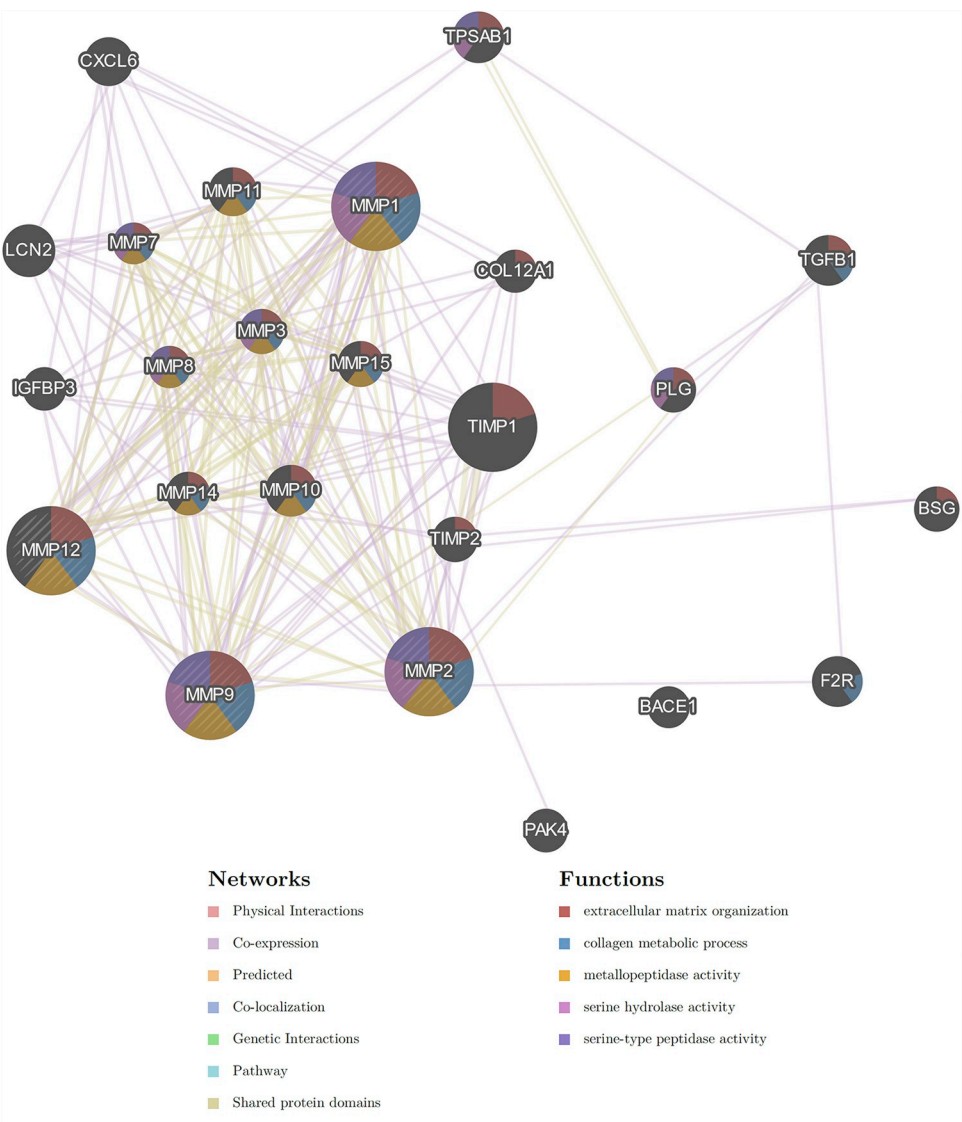

**Fig 1. Network integration of genes by GeneMANIA for MMP1, MMP2, MMP9 and MMP12.** The color of the lines that connect the genes indicates interaction types, including physical interactions, co-expression, predicted, co-localization, genetic interactions, pathway and sharing protein domains. The default color of the circle of the gene is grey. Red, blue, orange, pink and violet colors in the circle of genes, indicate the proteins that are encoded by the genes, and are contributing to biological function or signaling pathway.

S = 1.145) and $\geq$20 pack years (cOR = 6.29; 95% CI, 2.31–17.14, p<0.001, RERI = 0.362; AP = 0.058, S = 1.074), significantly increases the risk of COPD.

## Discussion

With this study, we estimated the SNP-SNP interactions of *MMP*s with smoking in COPD risk. This is the first data for the polymorphisms of *MMP*s, their interactions and COPD risk in the Mongolian population. One of the main results of the present study was that rs652438 polymorphism of *MMP12* positively interacted with cigarette smoking-related factors and increased the risk of COPD. Recent studies have found various evidence of the relationship between cigarette smoking and MMP12 activity. Exposure to smoking causes protein leakage,

**Table 7. Possible interaction of smoking-related factors with rs652438 polymorphism of *MMP12* gene in COPD.**

| Risk factors | | Cases N = 146 | Controls N = 208 | cOR | 95%CI | P value | RERI | AP | S |
|---|---|---|---|---|---|---|---|---|---|
| Smoking related factor | Genotype | | | | | | | | |
| Pack years | MMP12 gene (rs652438) | | | | | | | | |
| <20 | Non T/T | 5 (0.034) | 29 (0.14) | 1 | - | - | 0.362 | 0.058 | 1.074 |
| | T/T | 42 (0.288) | 72 (0.346) | 3.38 | 1.22–9.41 | 0.019 | | | |
| ≥20 | Non T/T | 22 (0.151) | 36 (0.173) | 3.54 | 1.19–10.51 | 0.023 | | | |
| | T/T | 77 (0.527) | 71 (0.341) | 6.29 | 2.31–17.14 | <0.001 | | | |
| CPD (number) | MMP12 gene (rs652438) | | | | | | | | |
| <20 | Non T/T | 10 (0.069) | 45 (0.216) | 1 | - | - | 0.671 | 0.108 | 1.148 |
| | T/T | 61 (0.418) | 101 (0.486) | 2.72 | 1.27–5.78 | 0.009 | | | |
| ≥20 | Non T/T | 17 (0.116) | 20 (0.096) | 3.83 | 1.49–9.81 | 0.005 | | | |
| | T/T | 58 (0.397) | 42 (0.202) | 6.21 | 2.81–13.72 | <0.001 | | | |
| ND | MMP12 gene (rs652438) | | | | | | | | |
| Low dependence | Non T/T | 9 (0.062) | 37 (0.178) | 1 | - | - | 0.448 | 0.099 | 1.145 |
| | T/T | 46 (0.315) | 77 (0.37) | 2.46 | 1.09–5.55 | 0.031 | | | |
| High dependence | Non T/T | 18 (0.123) | 28 (0.135) | 2.64 | 1.03–6.76 | 0.042 | | | |
| | T/T | 73 (0.5) | 66 (0.317) | 4.55 | 2.04–10.13 | <0.001 | | | |

CPD, Cigarette per day; ND, Nicotine dependence; cOR, Crude odd's ratio; CI, confidence interval; RERI, relative excess risk due to interaction; AP, attributable proportion; S, synergy index. The values are given as numbers (frequency). Odd's ratio and confidence interval were calculated by logistic regression.

such as plasminogen and prothrombin from serum to alveolar spaces. Those are converted into active form and it affects the secretion and activation of MMP12 [34, 35]. Another relation between cigarette smoke and MMP12 upregulation was reported by Botelho et al., who found cigarette smoke leads to an increase in the number of neutrophils and the level of granulocyte macrophage colony-stimulating factor (GM-CSF) in the lung. Also, mRNA expression of MMP12 was significantly higher in the cigarette smoking group with no anti-GM-CSF compared with anti-GM-CSF [36]. These reports suggest cigarette smoking increases GM-CSF, plasmin and thrombin in airway tissue, it upregulates MMP12 secretion and activation. Many studies reported the overexpression and increased activity on MMP12 linked to the development of emphysema and COPD [37–40]. Agne Babusyte et al., found the number of MMP12 expressing macrophages in bronchoalveolar lavage and induced sputum, was higher in smokers with COPD compared with COPD non-smokers and healthy controls [9]. Similarly, Alexandra Noël et al., reported second-hand smoke exposure upregulates MMP12 expression and activity [8]. Hautamaki et al., found that *MMP12*$^{-/-}$ mice were resistant to cigarette smoke induced emphysema. Interestingly, *MMP12*–/– mice also failed to accumulate macrophages in the bronchoalveolar lavage fluid [40, 41]. Also, *MMP12* knockout mice are markedly protected against airway remodeling. MMP12 has a wide range of degrading substrates including type III, IV and V collagen, elastin, proteoglycans and gelatin, which are the extracellular components of the airway. Besides degrading matrix, previous reports suggest MMP12 have a pro-inflammatory effect, associated with an increased level of active TNF-α and the secretion of neutrophils and macrophages in lung tissue [42, 43]. Andrew Churg et al. in 2003 and 2012, found during the smoke-induced inflammation the level of whole-lung E-selectin, endothelial activation marker and TNF-α were increased in only *MMP12* +/+ mice. Also, TNF-α overexpresser mice developed emphysema with increased expression of MMP12 and the pro-inflammatory chemokines [44]. It suggests that during smoke-induced inflammation MMP12 can act as a TNF-α converting enzyme (TACE), to release and increases active TNF-α then

drives pro-inflammatory reactions [43, 45]. On the other hand, previous studies have reported that the T allele of rs652438 polymorphism is associated with increased activity of MMP12 and COPD risk. In 2005, rs652438 T/T and T/C genotypes were reported as risk factors for smoking-induced COPD among Han nationality in northern China [46]. Imran Haq et al., reported that MMP12 activity was 3.45-fold higher for the T allele compared with the C allele and the number of macrophages was significantly greater in the airway sample from T/T carriers [21]. Ladina Joos et al., found that rs652438 polymorphism was associated with the rate of decline of lung function [47]. Xiao-Ling Yu et al., reported a meta-analysis and they found that rs652438 polymorphism was associated with a higher risk of COPD [48]. Also, the A-A haplotype of rs652438 and rs2276109 was associated with increased risk for severe and very severe stages of COPD [49]. These reports and our finding supports that cigarette smoking and the T allele, T/T genotype of rs652438 *MMP12* are positively interacting and associated with an increased count of MMP12 expressing macrophages and overexpression of MMP12 in airway tissue, then it may lead to develop COPD and emphysema.

Another important finding of the present study is the interaction of two SNPs located on *MMP2* and *MMP12* was detected as more than additivity, strongly increasing the risk of COPD susceptibility. Previously, little evidence was found about the association *MMP2* rs243864 with COPD [46, 48, 49]. Jazmín Hernández-Montoya et al., found that rs243864 G/G genotype of *MMP2* is strongly associated with increased risk of COPD (OR = 7.44) and it was the only study that evaluated the association between rs243864 and COPD risk [25]. Anna Vasku et al., reported that rs243864 SNP is located at promotor site of the MMP2 gene and three transcriptional factors can be bind with only T allele of rs243864 but not with G allele [50]. One of those transcriptional factors, which is the gut-enriched Krüppel-like factor (KLF4) and Bin Li et al., found KLF4 downregulates MMP2 expression [51]. This means in the presence of rs243864 T allele, KLF4 inhibits MMP2 expression. However in the presence of the G allele, MMP2 expression is not be inhibited by KLF4 and this may lead to overexpression of MMP2 to increase the risk of COPD. This explains why the G/G genotype appeared to have a strong risk for COPD in the present and Jazmín Hernández-Montoya's studies. Previously conducted studies showed rs243864 and rs652438 polymorphisms associated with increased risk of COPD, respectively but there are no genetic association studies that explored the interaction between them for COPD risk. To our knowledge, we found the individual who carried a G/G+T/T combination of rs243864 and rs652438, had a 12.92-fold (95% CI, 1.46–114.4) higher risk for COPD in the Mongolian population. In addition, the gene-gene interaction analysis showed a high degree of positive interaction (RERI = 6.94; AP = 0.537; S = 2.395) and linkage through co-expression (weight: 8.01%), shared protein domains (weight: 0.6%) between MMP2 and MMP12. Several hypotheses regarding the co-expression of MMP2 and MMP12 are possible. One of them is TGF-β signaling pathway, which is plays important role in MMPs expression. Krstic J et al mentioned that TGF-β signaling is regulates MMPs expression via activation of Smad and non-Smad signaling pathways [52]. Also, in the experiment by Lianyong Jiang et al., mRNA expression and protein level of MMP2 and MMP12 significantly increased in TGF-β1 treated cells compared with control group. Interestingly, after treated with siRNA that linc00511, the mRNA and protein levels are downregulated [53]. Based on the pathway database of Reactome and Genemania, both MMP2 and MMP12 proteins are involved in the biological processes, such as collagen type IV degradation, elastin, fibrillin 1, 2, 3 and aggrecan degradation. The above-mentioned findings support the fact that TGF-β and other signaling pathways can regulate co-expression of MMP2 and MMP12. Moreover, MMP2 and MMP12 are involved in the same biological processes. The overexpression of both MMPs can increase the risk of COPD. Further investigations will be required for the interaction between MMP2 and MMP12. We explored a few chosen genetic polymorphisms in small

sample sizes was the limitation we had. However, these findings might be applicable data for further studies.

## Conclusion

Our result supports the SNP-SNP and SNP-environment positive interactions between *MMP2* rs243864, *MMP12* rs652438 and smoking-related factors, strongly associated with increased risk of COPD.

## Supporting information

**S1 Table. The primers, restriction enzymes and length of fragments.**
(DOCX)

## Author Contributions

**Conceptualization:** Chimedlkhamsuren Ganbold, Jambaldorj Jamiyansuren, Ichinnorov Dashtseren, Sarantuya Jav.

**Data curation:** Chimedlkhamsuren Ganbold, Ichinnorov Dashtseren.

**Formal analysis:** Chimedlkhamsuren Ganbold, Enkhbileg Munkhzorig.

**Funding acquisition:** Sarantuya Jav.

**Investigation:** Chimedlkhamsuren Ganbold, Jambaldorj Jamiyansuren.

**Methodology:** Chimedlkhamsuren Ganbold, Enkhbileg Munkhzorig.

**Project administration:** Sarantuya Jav.

**Resources:** Chimedlkhamsuren Ganbold, Enkhbileg Munkhzorig, Ichinnorov Dashtseren.

**Software:** Chimedlkhamsuren Ganbold, Sarantuya Jav.

**Supervision:** Jambaldorj Jamiyansuren, Sarantuya Jav.

**Validation:** Chimedlkhamsuren Ganbold, Ichinnorov Dashtseren, Sarantuya Jav.

**Visualization:** Chimedlkhamsuren Ganbold.

**Writing – original draft:** Chimedlkhamsuren Ganbold, Jambaldorj Jamiyansuren.

**Writing – review & editing:** Jambaldorj Jamiyansuren, Sarantuya Jav.

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
