## [Decision Letter · Decision Letter 0]

31 Jan 2024

PONE-D-23-35781SNP-SNP positive interaction between MMP2 and MMP12 increases the risk of COPDPLOS ONE

Dear Dr. Jamiyansuren,

Thank you for submitting your manuscript to PLOS ONE. After careful consideration, we feel that it has merit but does not fully meet PLOS ONE’s publication criteria as it currently stands. Therefore, we invite you to submit a revised version of the manuscript that addresses the points raised during the review process.

We look forward to receiving your revised manuscript.

Kind regards,

Mohd Hussain Shah, PhD.

Guest Editor

PLOS ONE

Journal Requirements:

2. Thank you for stating the following financial disclosure:"This research received assistance from the Department of Science and Technology, Mongolian National University of Medical Sciences (Grant No.1/18), and the Mongolian Foundation for Science and Technology, Ministry of Science and Education, Mongolia (Grant no. ШуСс-2020/42)." 

3. Thank you for stating the following in the Acknowledgments Section of your manuscript: "This research received assistance from the Department of Science and Technology, Mongolian National University of Medical Sciences (Grant No.1/18), and the Mongolian Foundation for Science and Technology, Ministry of Science and Education, Mongolia (Grant no. ШуСс-2020/42)." 

Please remove any funding-related text from the manuscript and let us know how you would like to update your Funding Statement. Currently, your Funding Statement reads as follows: "This research received assistance from the Department of Science and Technology, Mongolian National University of Medical Sciences (Grant No.1/18), and the Mongolian Foundation for Science and Technology, Ministry of Science and Education, Mongolia (Grant no. ШуСс-2020/42)." 

Additional Editor Comments :

Hello,

Please go through the reviewers comments as below raised by the reviewer and answer them accordingly.

Ganbold et al. reported the SNP-SNP and SNP-environment positive interactions between MMP2 rs243864, MMP12 rs652438 associated with increased risk of COPD among Mongolian population. Moreover, smoking-related factors were also evaluated. It provides evidence of distinct populations and underscores the potential value of clinical intervention. However, there are still some issues that need clarification.

Major:

1. Was all p-value adjusted after multiple comparison? For example, Table 3, Table 4, and Table 5.

2. Page 9. Line153. Only recessive model was remaining after comparing the genotype frequency. I am confused about Table 4. The genotype frequencies of each SNP need to be calculated for both the COPD and control groups.

Minor:

1. Table 5. After adding rs1799750, rs11646643, and rs3918253, there is no significant difference. Do all three of these SNPs collectively exhibit a risk-resistant factor for COPD?

2, Page 11, line 169, “we found positive interaction … had increased risk for COPD” The interaction mechanism of rs243864 in MMP2 and rs652438 in MMP12 should be discussed.

Thank you.

Reviewers' comments:

Reviewer's Responses to Questions

**Comments to the Author**

1. Is the manuscript technically sound, and do the data support the conclusions?

Reviewer #1: Yes

2. Has the statistical analysis been performed appropriately and rigorously? 

Reviewer #1: I Don't Know

3. Have the authors made all data underlying the findings in their manuscript fully available?

Reviewer #1: Yes

4. Is the manuscript presented in an intelligible fashion and written in standard English?

Reviewer #1: Yes

5. Review Comments to the Author

Reviewer #1: Ganbold et al. reported the SNP-SNP and SNP-environment positive interactions between MMP2 rs243864, MMP12 rs652438 associated with increased risk of COPD among Mongolian population. Moreover, smoking-related factors were also evaluated. It provides evidence of distinct populations and underscores the potential value of clinical intervention. However, there are still some issues that need clarification.

Major:

1. Was all p-value adjusted after multiple comparison? For example, Table 3, Table 4, and Table 5.

2. Page 9. Line153. Only recessive model was remaining after comparing the genotype frequency. I am confused about Table 4. The genotype frequencies of each SNP need to be calculated for both the COPD and control groups.

Minor:

1. Table 5. After adding rs1799750, rs11646643, and rs3918253, there is no significant difference. Do all three of these SNPs collectively exhibit a risk-resistant factor for COPD?

2, Page 11, line 169, “we found positive interaction … had increased risk for COPD” The interaction mechanism of rs243864 in MMP2 and rs652438 in MMP12 should be discussed.

6. PLOS authors have the option to publish the peer review history of their article (what does this mean?). If published, this will include your full peer review and any attached files.

Reviewer #1: No

---

## [Author Response · Author response to Decision Letter 0]

15 Mar 2024

Dear Reviewer, 

Here, we have provided responses to the reviewer's inquiry below. 

We sincerely appreciate the reviewer's valuable feedback and the opportunity to address their inquiries. Your insights have contributed significantly to the improvement of our work. We have attached this response to reviewers file into system too.

Thank you again. 

Major:

1.Was all p-value adjusted after multiple comparison? For example, Table 3, Table 4, and Table 5.

Response:

For table 3, 5 all the p-values are not for adjusted. There are only Crude ORs and its p-value calculated by logistic regression. We revised the note below the table 3 and 5, please check. Thank you.

For table 4, we calculated both crude and adjusted ORs by logistic regression. Also, p-values were calculated for both ORs. We revised the note below the table 4, please check. Thank you again.

2.Page 9. Line153. Only recessive model was remaining after comparing the genotype frequency. I am confused about Table 4. The genotype frequencies of each SNP need to be calculated for both the COPD and control groups.

Response:

The genotype frequencies of each SNP calculated and compared for both groups with all models (recessive, dominant, multiplicative and over-dominant). However, we only included the results of selected models for each SNPs in Table 4. We used the computational and selection method for genetic model described by Horita and Kaneko. (Please see statistical analysis section.) The method calculates OR1 and OR2 based on genotype frequencies in both groups. The model was selected by intersecting the calculated OR1 and OR2 as shown in the graph below. The selected genetic model can be different for each SNPs but in this time, it was recessive for SNPs. We revised the note below the table 4, please check. Thank you again.

Minor:

1.Table 5. After adding rs1799750, rs11646643, and rs3918253, there is no significant difference. Do all three of these SNPs collectively exhibit a risk-resistant factor for COPD?

Response:

After adding SNPs rs1799750, rs11646643, and rs3918253, odd`s ratio of four SNP model was increased up to 3.14, but it was not significant (p>0.05). We believe that it`s because of the small sample size. Thank you again.

2.Page 11, line 169, “we found positive interaction … had increased risk for COPD” The interaction mechanism of rs243864 in MMP2 and rs652438 in MMP12 should be discussed.

Response:

We included some additional information related to MMP2, MMP12 and their relation in the discussion section. Please check the discussion section (line 242-249 and 256-267 of revised manuscript). Thank you again.

---

## [Editor Report · Decision Letter 1]

24 Mar 2024

Dear Dr. Jambaldori,

We’re pleased to inform you that your manuscript has been judged scientifically suitable for publication and will be formally accepted for publication once it meets all outstanding technical requirements.

The authors have answered most of the questions raised by the reviewers that improved the quality of this manuscript. 

Kind regards,

Mohd Hussain Shah, PhD.

Guest Editor

PLOS ONE

---

## [Editor Report · Acceptance letter]

9 May 2024

PONE-D-23-35781R1 

PLOS ONE

Dear Dr. Jamiyansuren, 

I'm pleased to inform you that your manuscript has been deemed suitable for publication in PLOS ONE. Congratulations! Your manuscript is now being handed over to our production team.

Kind regards, 

on behalf of

Dr. Mohd Hussain Shah 

Guest Editor

PLOS ONE